# Alternatives Therapeutic Approaches to Conventional Antibiotics: Advantages, Limitations and Potential Application in Medicine

**DOI:** 10.3390/antibiotics11121826

**Published:** 2022-12-16

**Authors:** Hiba Alaoui Mdarhri, Rachid Benmessaoud, Houda Yacoubi, Lina Seffar, Houda Guennouni Assimi, Mouhsine Hamam, Rihabe Boussettine, Najoie Filali-Ansari, Fatima Azzahra Lahlou, Idrissa Diawara, Moulay Mustapha Ennaji, Mohamed Kettani-Halabi

**Affiliations:** 1Faculty of Medicine, Mohammed VI University of Health Sciences (UM6SS), Casablanca 82 403, Morocco; 2National Reference Laboratory, Mohammed VI University of Health Sciences (UM6SS), Casablanca 82 403, Morocco; 3Laboratory of Virology, Oncology, Biosciences, Environment and New Energies, Faculty of Sciences and Techniques Mohammedia, University Hassan II of Casablanca, Casablanca 28 806, Morocco; 4Department of Biological Engineering, Higher Institute of Bioscience and Biotechnology, Mohammed VI University of Health Sciences (UM6SS), Casablanca 82 403, Morocco

**Keywords:** antimicrobial resistance (AMR), multidrug-resistant (MDR) bacteria, combination therapy, therapeutic strategies, infectious diseases

## Abstract

Resistance to antimicrobials and particularly multidrug resistance is one of the greatest challenges in the health system nowadays. The continual increase in the rates of antimicrobial resistance worldwide boosted by the ongoing COVID-19 pandemic poses a major public health threat. Different approaches have been employed to minimize the effect of resistance and control this threat, but the question still lingers as to their safety and efficiency. In this context, new anti-infectious approaches against multidrug resistance are being examined. Use of new antibiotics and their combination with new β-lactamase inhibitors, phage therapy, antimicrobial peptides, nanoparticles, and antisense antimicrobial therapeutics are considered as one such promising approach for overcoming bacterial resistance. In this review, we provide insights into these emerging alternative therapies that are currently being evaluated and which may be developed in the future to break the progression of antimicrobial resistance. We focus on their advantages and limitations and potential application in medicine. We further highlight the importance of the combination therapy approach, wherein two or more therapies are used in combination in order to more effectively combat infectious disease and increasing access to quality healthcare. These advances could give an alternate solution to overcome antimicrobial drug resistance. We eventually hope to provide useful information for clinicians who are seeking solutions to the problems caused by antimicrobial resistance.

## 1. Introduction

Multiple drug resistance (MDR) is identified by the World Health Organization (WHO) as one of the most serious threats to global health, food security and development [1]. It can affect anyone, at any age and in any country. It is now a major global public health challenge that arises for multiple reasons, including overpopulation, increased global migration, and selective pressure from increased use of antibiotics. The WHO has listed antibiotic resistance as one of the three most important public health threats in the 21st century (Figure 1) [2]. It estimates that infections caused by multidrug-resistant (MDR) bacteria (bacteria that are simultaneously resistant to three or more kinds of antibiotics used in a clinic) kill about 700,000 people worldwide each year, and that this number might rise to 10 million fatalities by 2050, exceeding the current yearly number of cancer-related deaths, if no action is taken [3,4,5].

This calls for the scientific community to design new antibiotics or innovative therapeutic approaches for treating critical priority antibiotic-resistant infections [5]. Common bacterial pathogens, such as *Klebsiella pneumoniae*, *Acinetobacter baumannii*, *Pseudomonas aeruginosa*, *Escherichia coli*, etc. have evolved and become resistant to multiple antibiotics, and their treatment is now becoming problematic (Figure 1). An increasing number of infections, such as pneumonia, tuberculosis, gonorrhea, or salmonellosis are becoming more difficult to treat as the antibiotics used to treat these infections lose their effectiveness. Unfortunately, the inadequate and irregular administration of antibiotics also contributes significantly to the development of antibiotic resistance, which leads to prolonged hospitalizations and increased medical expenses [7]. Additionally, several studies have reported that widespread use of antibiotics for hospitalized COVID-19 patients without established secondary infection was markedly increased, thereby leading to an increase in antimicrobial resistance through driving selection of MDR organisms [8,9,10,11]. The Centers for Disease Control and Prevention (CDC) in its special report of the year 2022 entitled “COVID-19: U.S. Impact on Antimicrobial Resistance” also concluded that the threat of antimicrobial-resistant infections is not only still present but has worsened [12]. Therefore, there is an urgent need for new classes of antimicrobials and other innovative approaches to fight against the emergence of MDR bacteria and escape the therapeutic impasse. In addition to traditional approaches, several new approaches (Figure 2), such as bacteriophages, antimicrobial peptides, essential oils and host-oriented therapies show great potential.

The objective of this review of the literature is to take stock of these different therapeutic approaches carried out over the last decade and to discuss their applications in the fight against the emergence of bacterial resistance to antibiotics. We also highlight the underlying mechanisms, advantages and limits of these mentioned antimicrobial strategies. Finally, we formulate a perspective and give our recommendations on potential practical directions and new antimicrobial strategies based on a brief conclusion.

## 2. New Antibiotics Therapy

The enactment of the 21st Century Cures Act and the Generating Antibiotic Incentives Now (GAIN) ACT, which resulted in the Qualified Infectious Disease Product (QIDP) indication, has rekindled innovation to manage antibiotic resistance. Plazomicin, Cefiderocol, Eravacycline and New β-Lactam– β-Lactamase Inhibitor Combinations are examples of effective QIDP antimicrobials [13,14].

### 2.1. Plazomicin

Plazomicin is a novel semisynthetic aminoglycoside antimicrobial derived from sisomicin to which a N1 2(S)-hydroxy aminobutyryl and a hydroxethyl group is added to the 6′ position [15]. It has been developed to target MDR *Enterobacteriaceae* including organisms capable of producing aminoglycoside-modifying enzymes (AMEs), extended spectrum beta-lactamases (ESBLs), and carbapenemases [16]. These resistant pathogens can be responsible of serious bacterial infections, including nosocomial pneumonia or bacteremia, which have become problematic throughout the world and to whom older aminoglycosides, including amikacin, gentamicin, and tobramycin, have limited activity against [16]. Plazomicin is a cationic, hydrophilic molecule which has low antibacterial efficacy in anaerobic conditions, such as an abscess or acidic urine [17]. Comparing plazomicin to sisomicin and gentamicin, blocking substituents cause a slight loss in antibacterial efficacy, while the presence of AMEs boost its activity against bacterial strains capable of producing AMEs [18,19]. In fact, with the exception of *Proteus mirabilis* and *Morganella morganii*, plazomicin was found to be more effective than the other investigated aminoglycosides against ESBL-producing *Escherichia coli*, ESBL-producing *Klebsiella pneumoniae*, carbapenem-resistant *Enterobacteriaceae* (CRE), and colistin-resistant *Enterobacteriaceae*. It also performed similarly to meropenem–vaborbactam and avibactam–ceftazidime combinations [16,20,21], and had similar activity to other aminoglycosides against Gram-positive isolates; which granted its Food and Drug Administration (FDA) approval for the treatment of adults with complicated urinary tract infections (cUTIs) and pyelonephritis caused by susceptible microorganisms and maintains bactericidal activity against most aminoglycoside-resistant *Enterobacteriaceae* [22,23].

As with other aminoglycoside antimicrobials, plazomicin is poorly absorbed and must be administered parenterally [15]. Monitoring renal function is a priority when utilizing this drug. Plazomicin was also found to penetrate into non-inflamed lungs to a similar degree as amikacin [13]. It should be noted that the FDA approved plazomicin with a black box warning for aminoglycoside class effects (nephrotoxicity, ototoxicity, neuromuscular blockade, and pregnancy risk) as it has for other aminoglycosides [24]. The FDA package insert recommends alternate dosing regimens of 10 mg/kg once daily in patients with CLCr ≤ 30 and <60 mL/min and 10 mg/kg every 48 h in patients with CLCr ≤ 15 and <30 mL/min [25]. Moreover, plazomicin has been evaluated in synergy experiments against MDR *Enterobacteriaceae*, including isolates with resistance to aminoglycosides and β-lactams [26]. The high acquisition cost of this aminoglycoside along with the cost of therapeutic monitoring will necessitate diligent antimicrobial stewardship, and be an issue for medical care providers [25]. Checkerboard experiments and time–kill assays both showed that plazomicin and piperacillin/tazobactam or ceftazidime worked together synergistically without any evident hostility. This next-generation aminoglycoside may also be used in combination therapy for severe Gram-negative infections caused by MDR *Enterobacteriaceae*. It has been evaluated with other antibiotics, mostly carbapenems against *Acinetobacter baumannii* [27]. Meropenem or imipenem in combination with plazomicin consistently resulted in synergy [22]. Newer beta-lactam/beta-lactamase inhibitors have demonstrated excellent activity against most major carbapenem-resistant phenotypes; yet, the emergence of resistance to ceftazidime/avibactam has already been reported, occurring both prior to exposure to the antibiotic and during active treatment [28].

Aminoglycosides have been utilized as add-on therapies with beta-lactams for serious infections for decades due to their synergistic mechanisms of action [25]. However, recent spread of resistance determinants against aminoglycosides has threatened this antimicrobial class. This is especially true in CRE isolates, which have been shown to harbor numerous AMEs phenotypes [25]. This could be another avenue for plazomicin to enter routine clinical use. It bears repeating that plazomicin did not receive an FDA indication for treating severe CRE infections; however, several data, both in vitro and in vivo, currently support its use in combination regimens for this indication [29,30]; as using plazomicin and meropenem or tigecycline appeared to be both more effective and safer than those using colistin [25].

### 2.2. Eravacycline

Eravacycline is a newly developed tetracycline derivative distinguished from earlier generations of tetracyclines and tigecycline in that it is a fully synthetic compound containing both a fluorine atom and a pyrrolidinoacetamido group side chain at the C9 position on its D-ring which protects against tetracycline-specific resistance mechanisms used by both Gram-positive and Gram-negative bacteria [31], including the “MP3” of *Morganella* spp., *Proteus* spp., *Pseudomonas* spp., and *Providencia* spp., that are naturally resistant to tetracyclines through a chromosomally mediated efflux pump, and many other isolates who developed resistance through genetic modifications [32]. In an in vitro surveillance study, eravacycline was compared with several other agents, including tigecycline, meropenem, and piperacillin-tazobactam, to evaluate the minimum inhibitory concentrations (MICs) for 50% (MIC50) and 90% (MIC90) of isolates. The organisms tested included: *Escherichia coli*, *Klebsiella pneumoniae*, *Acinetobacter baumannii*, *Staphylococcus aureus*, and *Enterococcus* [31]. For *Enterobacteriaceae* as a whole, Eravacycline had potent activity against many Gram-negative bacteria, even those with reduced susceptibility to tigecycline, with the MIC50 lower than the FDA breakpoint of 0.5. Compared with tetracycline and tigecycline (MIC50/90 8/>32 μg/mL and 0.5/4 μg/mL, respectively) eravacycline was more potent (MIC50/90 0.25/1 μg/mL) for *Acinetobacter* spp., including those demonstrating carbapenem, fluoroquinolone, and aminoglycoside resistance (MIC50/90 >8/>32μg/mL, 2/8 μg/mL, and 0.5/2 μg/mL for tetracycline, tigecycline, and eravacycline, respectively) [33]. In comparison to the eravacycline MIC90 of 0.13 μg/mL for MRSA, tigecycline had an MIC90 of 0.25 μg/mL, indicating a higher degree of susceptibility compared with the tigecycline breakpoint of 0.5 μg/mL 14. Against *Staphylococcus* spp.; both the MIC50 and MIC90 levels were lower than the FDA susceptibility breakpoint [33,34]. Moreover, eravacycline has demonstrated low MIC50 levels against some clinically significant anaerobic bacteria, including *Bacteroides fragilis*, *Clostridium difficile*, and *Clostridium perfringens* [33,35,36]. For *Neisseria gonorrhoeae* the MIC eravacycline was 0.12 μg/mL and MIC90 was 0.25 μg/mL, compared with 0.25 and 0.5 μg/mL, respectively, for tigecycline; no breakpoint has been defined for either agent [37]. Among isolates with reduced susceptibility to ceftriaxone or cefixime, 95% had an MIC that remained ≤0.25 μg/mL for eravacycline and ≤0.5 μg/mL for tigecycline. Among isolates with reduced susceptibility to azithromycin, the MIC remained ≤0.25 μg/mL for 87% tested against eravacycline and ≤0.5 μg/mL for 100% tested against tigecycline, and all isolates had MIC ≤ 0.5 μg/mL for both eravacycline and tigecycline [38]. Thus, with a broad spectrum of activity against *Enterobacteriaceae*, resistant Gram-positive organisms, and anaerobes, its primary use in therapy will most likely be for the treatment of infections caused by resistant pathogens in patients who cannot receive alternative agents. The case for this use is strengthened in areas with high rates of ESBL-producing bacteria and CRE [39].

Eravacycline may also be a desirable choice for people at risk for *Clostridium difficile* infection due to its in vitro efficacy against the bacterium, and in treating a variety of resistant Gram-negative and mixed infections where tigecycline would not be useful because of a comparatively improved tolerability and adverse effect profile in studies to date. Moreover, eravacycline’s popularity as an alternative therapy for particular populations is increased by the fact that it is not a β-lactam antibiotic [34,39,40]. In an effort to stop the spread of carbapenem resistance, it is also another option as a carbapenem-sparing regimen [41]. However, one novel tetracycline limitation is that it is only available as an intravenous infusion, as its oral formulation was discontinued after its disappointing failure and poor outcomes in clinical studies [42,43]. Whereas there are several oral options available for resistant Gram-positive infections, there are limited oral options for resistant Gram-negative infections. An oral eravacycline formulation could have greatly affected the treatment of these off-label infections, facilitated the removal of central intravenous lines, and potentially decreased the hospital length of stay. Warnings and precautions associated with eravacycline are those common among the tetracycline class, including tooth discoloration and reversible inhibition of bone growth which preclude its use beyond the first trimester of pregnancy and in children younger than 8 years [40].

### 2.3. Cefiderocol

Cefiderocol is a novel parenteral siderophore cephalosporin targeting Gram-negative bacteria, including strains with carbapenem resistance. The structural characteristics of cefiderocol; in addition to the similar chemical structures of both ceftazidime and cefepime, which are able to withstand hydrolysis by β-lactamases; show a unique chemical component which is a catechol moiety on the C-3 side chain that chelates iron and mimics naturally occurring siderophore molecules. Cefiderocol has demonstrated structural stability against hydrolysis by both serine- and metallo-β-lactamases (MBL), including clinically relevant carbapenemases such as *Klebsiella pneumoniae* carbapenemase, and oxacillin carbapenemase [44,45]. Given its unique siderophore mechanism and potent activity against several Gram-negative bacteria proven by several large multinational in vitro and in vivo studies, which was not demonstrated with earlier monobactam conjugates; it was approved by the U.S. FDA for the management of cUTIs, and is considered to be a viable option for several MDR infections where limited effective and well-tolerated antibiotics exist [45,46,47]. However, no clinically relevant in vitro activity against most Gram-positive and anaerobic bacteria has been demonstrated [44]. Similar to other β-lactam antibiotics, cefiderocol is generally well tolerated [48]. The standard dose of cefiderocol is 2 g administered every 8 h as a 3 h infusion with dose adjustments recommended for patients with a creatinine clearance of ≤60 mL/min and an increase in frequency to every 6 h for patients with augmented renal clearance (CLCR ≥ 120 mL/min) [48].

The most common adverse events reported in clinical trials were increases in serum alanine aminotransferase (ALT) and aspartate aminotransferase (AST) which requires periodic monitoring of liver enzymes in patients receiving cefiderocol therapy [49]. Concerns about adverse events related to iron homeostasis in humans have been discussed considering the unique mechanism of transport into bacterial cells. In three published clinical trials to date, anemia-related adverse events and variables related to iron homeostasis were similar between cefiderocol and comparator arms [50,51,52]. Resistance to cefiderocol is complex and not well characterized and estimates of the frequency of acquired resistance to cefiderocol are currently un-known.

### 2.4. New Combination Antibiotic Therapy

Numerous innovative β-lactam–β-lactamase inhibitor combinations (BLBLIs) have been created as a result of the limited arsenal against drug-resistant Gram-negative bacteria [53]. Clinical data regarding the use of these drugs to treat MDR bacteria are limited and rely mostly on non-randomized studies. These drugs provide various levels of in vitro coverage of CRE.

#### 2.4.1. Ceftazidime–Avibactam

Ceftazidime–avibactam (CAZ–AVI), a combination of the anti-pseudomonal third-generation cephalosporin ceftazidime that is hydrolyzed by class A ESBLs and carbapenemases, class B carbapenemases, and class C cephalosporinases but not by most class D carbapenemases [53]; and the novel β-lactamase inhibitor avibactam that inhibits class A, class C, and some of the class D β-lactamases providing, as published data have shown, a broad coverage of Gram-negative bacteria including highly resistant strains, such as ESBL-, AmpC-, and serine CPE and *Pseudomonas aeruginosa*, as well as some of the bacteria producing class D carbapenemases, such as OXA-24, OXA-40, OXA-69 (in *Acinetobacter baumannii*) and OXA-48 (in *Klebsiella pneumoniae*) but not against MBL producers [53,54]. The recommended administration and dosage regimen is 2 g of ceftazidime and 0.5 g of avibactam in continuous infusion administered over 2 h with a dosage of three times a day [55]. CAZ–AVI exhibits linear pharmacokinetics [56]. It does not undergo liver metabolism and is only weakly attached to proteins. Because it is renally excreted, dosages must be changed in cases of renal failure [57]. An observational, prospective, multicenter study that included 137 patients and isolates primarily from bacteremia (46%) and respiratory isolates (22%), of which 28% were treated with CAZ–AVI and 72% with colistin, was conducted to compare the two drugs’ effectiveness in the treatment of *Klebsiella pneumoniae*. Compared to patients treated with colistin, those who received CAZ–AVI had a 64% higher likelihood of a favorable outcome [56]. Patients with bacteremia caused by carbapenemase-producing *Klebsiella pneumoniae* had improved Sequential Organ Failure Assessment (SOFA) scores when CAZ–AVI was given as a salvage therapy [58]. Further, CAZ–AVI and aztreonam work together to overcome resistance brought on by enterobacteria’s synthesis of MBLs [59].

#### 2.4.2. Ceftolozane–Tazobactam

Ceftolozane–Tazobactam (C/T) is a new antibiotic resulting from the combination of a novel cephalosporin, structurally similar to ceftazidime, with tazobactam, a well-known β-lactamase inhibitor. This pairing illustrates the feasibility of combining a β-lactam and β-lactamase inhibitor that are not perfectly matched pharmacokinetically. In fact, they share similar protein binding values but differ in half-life and metabolic disposition [60]. They are well-tolerated, with the most frequent adverse events being those associated with any other cephalosporin, such as nausea, vomiting, and diarrhea. The association has shown activity against MDR *Pseudomonas aeruginosa* and ESBL-producing *Enterobacteriaceae* and has been recently approved for the treatment of cIAIs, with HABP/VABP, and cUTIs, including pyelonephritis, by the U.S. FDA and EMA. It has not yet been approved for use in pediatric patients [53]. However, the drug has special value for clinicians to prescribe in any kind of infectious localization and found to be valuable in suspected or documented severe infections due to MDR *Pseudomonas aeruginosa*. Additionally, it is a promising carbapenem-sparing agent that should be used thoughtfully for the treatment of infections caused by ESBL-producers, thus allowing a carbapenem-sparing strategy [60]. The physical compatibility of C/T with the other 95 common intravenous drugs has been examined in numerous studies [61,62,63]. C/T was compatible with 90.5% of the tested drugs, including metronidazole. It was incompatible with albumin, amphotericin B (both deoxycholate and lipid formulations), caspofungin, cyclosporin, nicardipine, phenytoin and propofol [53]. While C/T has limited effectiveness against anaerobes, it covers a wide range of Gram-negative bacteria, including MDR and extensively drug-resistant (XDR) *Pseudomonas aeruginosa* and ESBL-producing *Enterobacteriaceae*. It is noteworthy that C/T has only sporadic or no activity against *Staphylococcus* spp.; *Enterococcus* spp.; *Acinetobacter* spp., *Clostridium difficile*, and other resistant pathogens (such as carbapenemase makers) [64,65,66,67].

## 3. Phage Therapeutics

Phage therapy dates back to the beginning of the 20th century, even before Alexander Fleming’s discovery of penicillin in 1928 [68]. The first phage activity dated back to 1896, when Ernest Hankin reported that the waters from the rivers Ganges and Yamuna in India possessed antibacterial activity against *Vibrio cholerae* [69]. In the late 1910s, and following initial work by the English bacteriologists Ernest Hankin and Frederick Twort, a French microbiologist from the Pasteur Institute (Felix d’Herelle, 1917) identified viruses that specifically and selectively parasitized bacteria and named them “bacterium eaters” (bacteriophages) [70,71]. It was d’Herelle who first developed the notion of using phages therapeutically to treat bacterial infection with encouraging results [69]. However, since the discovery and development of antibiotics, Phage therapy was largely abandoned in the Western world due to the efficacy and promise held by antibiotics, with the exception of the Soviet Union and some Eastern European countries [69]. Recently, and in the face of the rapid emergence of resistant bacteria, phages (estimated to exceed 1031 particles) have re-emerged as alternative and complementary therapies to control bacterial infections [72]. Thus, phage therapy has shown to be an interesting alternative in the fight against multi-drug resistant bacteria [73]. Phages or bacteriophages are lytic viruses that exclusively and specifically infect bacterial species, showing bactericidal effects against both Gram-positive and Gram-negative bacteria [74,75]. In contrast, some tend to be specific to a particular species or strain of bacteria [76]. The tailed double-stranded DNA phages (order Caudovirales) are the most studied group and are thought to account for 96% of all phages and are easily isolated from various environmental sources (soil, wastewater, and aquatic environments) [73]. By adhering, via tail proteins, to specific surface receptors of bacteria, phages insert their genetic material into their bacterial hosts [69]. Several types of life cycles can be triggered by bacteriophages, of which the two most frequent are the lytic and lysogenic cycles [68]. During the lysogenic cycle, the virion DNA is incorporated in the bacterial genome. The resulting prophage replicates its genetic material within the bacterial cell without damaging it until the lytic cycle is trigger [68,77]. Undoubtedly, prophages are supposed to be shunned and lytic phages are selected. During the lytic cycle, the phage uses the cellular machinery to yield as many as 20,000 new virions per infected bacterial cell in optimal conditions [68]. These phages secrete lytic enzymes (endolysins) that hydrolyze the bacterial cell wall to ensure phage release [77,78]. Since then, it has become clear that phage therapy and the application of its endolysins offer the possibility to apply more specific antibacterial treatments and propose a potential solution to the problem of antibiotic resistance [79]. The table below (Table 1) shows some advantages of applying phage therapy to fight bacterial infections.

### 3.1. Applications in Medicine

Before the dawn of antibiotics, phage therapy had been used to treat a broad range of bacterial infection diseases, including cholera [89,90,91], pediatric dysentery [92], bubonic plague [93], typhoid fever, skin and surgical site infections, peritonitis, septicemia, and external otitis [92,93]. However, in 1934, the failed attempts to reproduce positive findings had inspired the opposition from the Council on Pharmacy and Chemistry of the American Medical Association [91]. This opposition has not prevented parts of Eastern Europe (such as Georgia, Poland and Russia) from continuing to use phages in routine medical practice and today provide us with a rich source of empirical data [94]. For example, The Eliava Institute of Bacteriophage, Microbiology, and Virology in Georgia is one of the longest-running institutions where phage therapy has been provided to frequent bacterial diseases related to urology, pediatrics, internal medicine, and gynecology [71]. Recently, phage therapy has been re-employed in the United States and Europe, for the treatment of infections related to burn injuries or soft tissue and skin trauma, osteomyelitis, sepsis, bacteremia, and otitis media as well as urinary tract, pulmonary, and prosthetic device-associated infections, especially when mono- or multi-infected patients with multi-resistant bacteria are without effective treatment options or are terminally ill [71,95]. In Table 2 below, we quote the references of the main applications of phage therapy (as an adjuvant or alternative therapy to antibiotics) conducted on human patients infected with various types of MDR bacteria [2]. Results from these studies indicate that this therapy has immense potential with applications in human medicine.

### 3.2. Limitations

Although phage therapy has come a long way, and is considered a promising alternative to antimicrobial agents, they have a dark, poorly explored side (Table 3). These limitations complicate the design of clinical protocols, undermine confidence in phage application, and need to be cleared before establishing successful phage therapy on a global scale.

## 4. Antimicrobial Peptides

In recent years, epidemics and outbreaks have shown that public health may be threatened globally in terms of infectious diseases, so this problem urgently requires finding alternatives to traditional antibiotics that are novel and less prone to bacterial resistance. In the quest for new antibiotics, antimicrobial peptides (AMPs), also known as host defense peptides, have recently received a great deal of interest [146,147,148]. AMPs are a class of small peptides that exist widely in nature and are important components of the innate immune system of different organisms. They have a broad inhibitory effect on bacteria, fungi, parasites and viruses. They were discovered in 1939, following the discovery of lysozymes in 1922, when microbiologist Rene Dubos isolated a strain of Bacillus from soil, an antibacterial agent called gramicidin, which was shown to protect mice from pneumococcal infection [149]. Subsequently, several AMPs have been identified in both the prokaryotic and eukaryotic kingdoms [150]. The first animal-derived AMP described was defensin, which was isolated from rabbit leukocytes [151], subsequently lactoferrin was identified in cow’s milk, and human leukocyte lysosomes and the human female reproductive tract have been shown to contain low-molecular weight AMPs [152]. To date, more than 3000 AMPs have been discovered, characterized, and annotated in the AMP database (APD3) [153].

Current research focuses on these natural compounds as innovative anti-infective drugs and novel immunomodulators [148,154], but also in food, livestock, agriculture and aquaculture. Interest in AMPs has recently increased during the severe acute respiratory syndrome coronavirus 2 (SARS-CoV-2) pandemic in the search for new antiviral molecules to counter COVID-19 [154].

### 4.1. Advantages

Since the discovery of natural bioactive AMPs capable of broadly striking multiple pathogens, peptide-based therapeutics are under the scrutiny of researchers. AMPs inhibit a wide range of microorganisms through diverse and special mechanisms by primarily targeting cell membranes or specific intracellular components [155]. AMPs are potential multifunctional therapeutic agents, which are effective against a broad spectrum of microorganisms. They are called “natural antibiotics”. Some AMPs can cause rapid death of Gram-positive, Gram-negative, fungi, parasites, encapsulated viruses or tumor cells within minutes. AMPs have a low risk of resistance development and can even inhibit antibiotic-resistant microorganisms [155]. This principal stems from the fact that AMPs generally (but not always, as specified above) strike the lipid component of the plasma membrane, a cellular component that is inherently thought to be difficult to alter in its basic physicochemical characteristics by microbial targets. Finally, AMPs are increasingly seen as a promising therapeutic alternative for the treatment of biofilm-associated infections, one of the major threats in the field of bacterial infections. All these advantages make AMPs ideal candidates for pharmacological applications.

AMPs exert their antimicrobial effects primarily through two different mechanisms. Membrane-targeting AMPs alter the structural integrity of the cell membrane while AMPs that use non-membrane targeting mechanisms primarily inhibit the synthesis of nucleic acids, essential enzymes and other functional proteins. These novel proteins can be of natural or synthetic origin. AMPs from many species, including amphibians, insects, mammals, and fish, account for 75.65% of total AMPs, while the remainder are mainly from plants and bacteria and account for 13.5% and 8.53% of total AMPs, respectively [156]. Bacterial AMPs are often referred to as bacteriocins, effective at lower concentrations than AMPs and have a limited effect on a few species. Bacteriocins display antimicrobial activity and inhibit major microbial competitors [157,158]. Remarkably, resistance to antibiotics and bacteriocins are independent and no cross-resistance has been recorded so far [159]. The genes involved in bacteriocin production are a part of mobile genetic elements such as conjugative transposons (nisin) and plasmids (lactococcin 972) [160]. This facilitates their propagation and explains, for example, the ability of *P. pentosaceus* and *Lactobacillus* to synthesize PA1/AcH pedicin [161]. It can be concluded that bacteriocins are a novel perspective to overcome the resistance problem provided that the mechanism function of these molecules and their degradation be thoroughly understood. In turn, the marine environment is known to be one of the richest sources of AMPs, including the oceans that cover just over 70% of the Earth. Unlike the terrestrial environment, they are more adaptable to harsh environmental conditions, such as high salinity [156,162]. Plant-derived AMPs are peptides that exhibit strong and broad-spectrum antimicrobial activity. In addition to the strong microbicidal activity of plant AMPs against viruses, bacteria, fungi, parasites, and protozoa, they also have anti-insect activity against oomycetes and herbivore pests, and anticancer activity against some cancer cells [155,156,162,163]. Regarding insect AMPs, they play an important role in the humoral immune system. Insect AMPs are synthesized in the body fat of an insect and stored in the hemolymph [164]. Over 200 AMPs have been identified in insects to date.

Despite the diversity of the many advantages of AMPs extracted from natural sources, they have a number of disadvantages, including poor stability, salt tolerance, and high toxicity, which hinder their widespread therapeutic use, hence the emergence of synthetic AMPs. Several methods have been developed to design new synthetic AMPs by modifying the sequences of AMPs naturally present in various organisms. It has been shown that small changes in amino acid composition can result in changes in all conformational and physicochemical properties of a peptide [165]. All in all, these AMPs are safe, with no or fewer toxic side effects, and difficult to induce bacterial drug resistance compared to conventional antibiotics [166].

### 4.2. Applications in Medicine

AMPs are multifunctional agents with several therapeutic functions such as anti-inflammatory, immunomodulatory, endotoxin neutralization activities, and cytotoxic effects on cancer cells, making them good candidates for pharmacological practices, in addition to their direct antimicrobial effects [167,168]. Rapid and broad-spectrum activities, versatile use opportunities, and low potentials for resistance development of AMPs are the main factors increasing their attractiveness in the biopharmaceutical industry and investment in the peptide antibiotics market. Several studies have reported the relevance of nisin in the treatment of several infectious diseases, such as mastitis, oral, respiratory, and skin [148,152,169]. Mastitis is a common inflammatory disease in lactating women that results in cessation of lactation [170]. *Staphylococcus aureus* and *Staphylococcus epidermidis* are two common agents that cause infections associated with mastitis [170]. The peptide nisin causes inhibition of bacterial growth through the formation of membrane pores and by interrupting cell wall biosynthesis through a specific lipid II interaction [171].

Another example of bacterially derived AMPs used clinically as an alternative to antibiotics is gramicidin, which is a mixture of gramicidin A, B, and C. These are AMPs naturally produced by *Brevibacillus brevis*, with activity against several Gram-positive bacteria, inducing membrane depolarization and consequently cell lysis [152,172,173]. Gramicidin is a constituent of Neosporin^®^, a triple antibiotic used in ophthalmic and topical preparations [152]. It has been approved and marketed as an anionic AMP for the treatment of skin infections caused by Gram-positive bacteria [174]. A recent study demonstrated that pretreatment with Tachyplesin III on mice protects them against *Pseudomonas aeruginosa* and *Acinetobacter baumannii* infection, reduces the production of pro-inflammatory cytokines (IL-1β, IL-6 and TNF-α) and induces macrophage phagocytosis, fundamental to exert bacterial clearance in a dose-dependent manner [152,175]. All of these findings need to be confirmed in human clinical trials. Given the wide variety of AMPs existing in nature, it is expected that other new pharmacologically active peptides inspired by nature may find clinical applications in the future.

### 4.3. Limitations

Although AMPs have many benefits, only three AMPs have been approved by the U.S. FDA for therapeutic use, namely, gramicidin, colistin, and daptomycin [176]. Gramicidin is active against a range of Gram-positive and Gram-negative bacteria, although its severe toxicity to human erythrocytes has a clinical indication limited to topical applications. Polymyxin and colistin, which are cationic peptides that have been in use for decades, have seen a resurgence of interest in recent times due to their strong activity against MDR Gram-negative pathogens. Their ability to bind the lipid A component of LPS makes them valuable weapons, the last resources to fight septic shock, notwithstanding their known nephrotoxicity. However, resistance has emerged and is spreading at an alarming rate, jeopardizing the efficacy of these valuable therapeutics. As for daptomycin, this membrane-bound cyclic lipopeptide was given the green light by the FDA in 2003 to treat Gram-positive infections. In recent years, *Staphylococcus aureus* resistance has been increasingly reported and the search for substitutes that can extend the clinical use of this important antibiotic is actively underway. In addition to the AMP resistance described above, there is a level of difficulty in bringing these molecules to market, whether for topical or systemic treatment. Polyphor’s phase 3 clinical trials of the safety and efficacy of intravenous murepavadin were stopped prematurely due to elevated serum creatinine levels in AMP-treated patients, indicating renal failure [177].

## 5. Nanoparticles

The first traces of the empirical use of nanoparticles by mankind dates back to the Mayan empire (in 800 BC), they used a blue dye able to resist weather conditions made from a mixture of nanoporous clay and dye containing metallic nanoparticles [178]. The production of nanoparticles has been a fast-growing subject for biomedical applications, particularly in the treatment of tumors and their diagnostics, since the beginning of the 1970s and exponentially since the 2000s [179]. These nano-objects’ outstanding features (optical, physical, thermal, chemical, etc.) enable their usage as vectors, as well as for diagnostic and therapeutic reasons. Nanotechnologies dedicated to medical applications are defined today under the term “nanomedicine”. The nanoparticles with antimicrobial activity are inorganic particles of nanometric size. They can attack the membrane of bacteria or yeast and penetrate these pathogens to interact with a specific target, but also induce the production of free radicals. More importantly, the multiple and unique mechanisms of action of nanoparticles make it unlikely that bacteria can develop resistance, and may aid in limiting the global crisis of emerging bacterial resistance [179,180].

### 5.1. Advantages

Nanomedicine is a drug delivery using nanovectors allowing the transport and release of the active ingredient at its pharmacological target. This approach, therefore, increases the efficacy of drugs and limits their adverse effects by modulating the pharmacokinetic pathway and bioavailability [181]. Unique antibacterial qualities are present in the metal nanoparticles (MNPs) made from silver, gold, zinc oxide and titanium oxide. Their chemical composition enables longer binding, active targeting of antibiotics with surface functionalization at the target location, and protection against enzymes degradation [182]. The most interesting part is that they can deliver antibiotics in the intracellular compartment where the pathogen is. Thus, they improve drug stabilization, increase drug circulation and provide improved delivery at the target and a higher treatment effectiveness. In parallel, they lessen toxicity and side effects, increase antibacterial spectrum, and decrease required medication dose and even the treatment time [183,184]. All these benefits make their use among the most promising strategies to overcome antibiotic drugs resistance [185].

### 5.2. Applications in Medicine

The nanobiotics are the next generation antibiotics and can be used as medication against bacterial infections with mainly intracellular bacteria. The antibacterial properties of noble metals (mainly copper and silver) have been known for a long time and are used in everyday life. For example, some hospitals have equipped their door handles with copper coatings in order to limit the transmission of bacteria in the hospital environment [186]. Studies of nanoparticles have shown in vitro antimicrobial activity against MDR organisms, including ESKAPE pathogens (*Enterococcus faecalis*, *Staphylococcus aureus*, *Klebsiella pneumoniae*, *Acinetobacter baumannii*, *Pseudomonas aeruginosa* and *Enterobacter*) [180,187,188,189]. In fact, a study clearly established the dose-dependent antibacterial activity of silver nanoparticles against enterobacteria like *Escherichia coli* or *Klebsiella pneumoniae* with an honorable MIC of about 1.4 µg/mL, as well as the antibacterial activity of gentamicin, cefotaxime, and meropenem towards these species was increased in the simultaneous presence of silver nanoparticles [190]. Other studies have demonstrated both in vitro and in vivo efficacy against Gram-positive and Gram-negative bacteria [191,192]. Furthermore, it could be an alternative in the treatment of tuberculosis, which is a major health problem infecting millions of people worldwide. Misuse of first-line drugs could lead to MDR tuberculosis, which is then treated with chemotherapy or second-line drugs such as ethionamide (ETH). However, patients often find it difficult to adhere to treatment regimens that require high doses of ETH. The nanoparticles could encapsulate synergistic drugs, which would greatly simplify treatment and increase patient compliance [181].

In the end, all of these studies remain at the experimental scale, in vitro or in vivo using animal models. Clinical translation requires a thorough understanding of the pharmacokinetics and pharmacodynamics of NPs and an evaluation of their toxicity at the cellular and systemic levels [180]. Therefore, the metal nanobiotics have been found to be more effective than the antibiotics alone due to their stability in the case of infections caused by intracellular germs [193,194,195]. In practice, the nanobiotics would provide drug delivery systems for systemic, oral, transdermal, or other application methods, better antibacterial wound dressing, excellent implantable medical device coatings and more effective antibacterial vaccine adjuvants [193].

### 5.3. Combination Therapy

The nanoparticles can be combined to antimicrobial drugs (antibiotics, antivirals, antifungals). The combination with antibiotic molecules is the most documented, and they are called the nanobiotics [196]. The MNPs are used for their reduced cost, easy synthesis and use [193]. The MNPs are based on different types of metal: mainly gold (Au), silver (Ag), copper oxide (CuO), iron oxide (Fe_3_O_4_ or Fe_2_O_3_), titanium dioxide (TiO_2_), and zinc oxide (ZnO) [193]. Silver nanoparticles (AgNPs) are used not only in research laboratories but also in the human and veterinary field due to their catalytic, unique optical and biological properties. Gold nanoparticles’ (AuNPs) major benefit is their simple production and modification by one or more chemically distinct thiol ligands and they are considered as the least toxic MNP [182]. Since MNPs operate as a drug carrier, they make it easier for antibiotics to be transported to the cell surface. This interaction between the nanoparticles and antibiotics raises the concentration of antimicrobial drugs at specific locations on the cell membrane. Particularly, the contacts with cells are improved and the permeability of the membranes are increased by AgNPs’ and AuNPs’ affinity for sulfur-containing proteins of bacterial cell membranes. This makes it easier for the antibiotics to enter the cell [194].

### 5.4. Limitations

Nanoparticles represent a risk to human health despite their excellent qualities and vast variety of applications. Studies conducted in vitro and in vivo have demonstrated that MNPs can enter cells and cause oxidative stress, inflammation, DNA damage, and organ toxicity, which restricts their use [197,198]. The main properties of MNPs responsible for their toxicological effect have been attributed to:Size, due to their rapid diffusion into human cells and their ability to pass across the blood-brain barrier (200 nm). NPs below 10 nm often exhibit substantial antibacterial activity but also high cytotoxicity;Agglomeration, which aids in the sedimentation process and slows NP diffusion which increases effective dosages;Surface charge, in order to control protein binding to NPs, cellular uptake, oxidative stress, autophagy, inflammation, and apoptosis, NPs’ charge is crucial (charged NPs have been shown to be more cytotoxic than neutral forms, and positively charged NPs were more cytotoxic than negative variants of similar size). Currently, MNPs can be designed to reduce their toxicity to humans [199]. The size can be tailored for optimal efficacy, and capping agents can be used to prevent agglomeration, avoid undesirable nanoparticles oxidation and enhance ion release. Commonly used capping agents are oleic acid, polyacrylic acid, polyethylene glycol (PEG), polyvinyl alcohol (PVA) and polyvinylpyrrolidone (PVP) [200,201].

## 6. Essential Oils

Natural products, such as medicinal plants, essential oils (EOs), and herbal extracts are regarded as promising alternative agents. EOs are among the most economically relevant plant-derived products, being frequently responsible for several health-promoting properties [202,203]. EOs are formed by the MAP as secondary metabolites, and are very heterogeneous mixtures that may contain dozens of compounds at different concentrations [204]. Presently, over 3000 EOs are known, 10% of which are commercially and economically relevant. These products are potential reservoirs of many bioactive compounds with several beneficial properties, and they are aligned with the current consumer preference for natural products. They also offer the advantage of being better tolerated in the human body with fewer side effects [205,206].

### 6.1. Applications of EOs

In nature, EOs play an important role in the protection of the plants, by acting as antibacterials, antivirals, antifungals, insecticides, and also against herbivores by reducing their appetite for such plants. They may also attract certain insects to favor the dispersion of pollens and seeds or repel others that are undesirable. Due to their bactericidal and fungicidal properties, they are also used as alternatives to synthetic chemical products to protect the ecological equilibrium without showing the same secondary effects [207,208]. EOs have been used for over 5000 years in various cultures and for a variety of different purposes, including personal care (perfumes and cosmetics), foods, home care, repellents for humans and animals (livestock and domestic animals), and health-promoting agents for the treatment of various diseases [203,209,210,211]. EOs are generally known to exert various pharmacological effects, such as antiallergic (lavender, sage), anticancer (salvia), anti-inflammatory (eucalyptus, lavender), analgesic (lavender), anxiolytic (common fennel), antidepressant (citron) immunomodulatory (eucalyptus) antibacterial (clove, cumin, thyme, oregano), antifungal (curcuma, lavender, eucalyptus), antiviral (eugenol, eucalyptus, ginger grass, clove), insecticidal (artemisia, eucalyptus, lavender), and antioxidant activity (coriander, lavender, aneth) [206,212,213,214]. Also, EOs could be used as alternative preservatives to increase the shelf lives of cereals and crops and in food safety [215]. Thus, the composition and pharmacological properties of EOs are ongoing subjects of extensive research interests.

### 6.2. Antimicrobial Effect of EOs

EOs and their major components have proven to be effective in controlling the spread of certain bacterial agents. The antibacterial properties of EOs have been known for a long time and today there are a good number of important publications have confirmed their bacteriostatic and bactericidal effects against pathogenic bacterial strains, even at very low concentrations [216,217]. Several molecules present in EOs are endowed with antibacterial properties, especially phenols (such as carvacrol, thymol and eugenol), alcohols (such as such as linalool) and aldehydes (such as cinnamaldehyde). The antibacterial effects are influenced by different factors such as the chemical composition of the EO tested, the experimental method used, and the bacterial strain tested. Their antibacterial action depends on the concentration of the major compounds, synergistic effects and/or additives and minor compounds present [218,219]. The bioactive components of EOs have specific antibacterial actions; for example, the cellular destruction of pathogens is a result of the ability of these compounds to disrupt the microorganism’s cell membrane, which results in changes in cell morphology, alterations in membrane permeability and leakage of electrolytes. For instance, thymol, carvacrol and eugenol (major phenolic compound of several EOs) can increase the amount of C16 and C18 saturated fatty acids and decrease the amount C18 unsaturated fatty acids. The action of EOs is not restricted to fatty acids themselves but can also affect the enzymes responsible for their biosynthesis. For example, some studies have shown that some EOs inhibit by different mechanisms (competitive, non-competitive and uncompetitive) [213,220]. The different components of EOs can act on the proteins present in bacteria and affect cell division. Furthermore, they can decrease the rate of ATP production and cause disturbances in the membrane respiratory chain. Some EOs have an action on quorum sensing, used by bacteria to coordinate and ensure communication between them [221,222]. Several studies have investigated the synergy between EOs components [223,224]. A variation between different outcomes was observed according to the test organism, antibiotic and essential oil tested [225]. The combination of two antimicrobial agents results in an antagonistic, indifferent, or synergistic effect, depending on the mechanisms exerted by each agent against the test organism, the characteristics of the test organism, and any chemical interactions between the two antimicrobial agents [226].

### 6.3. Limitations

Despite being natural compounds and their therapeutic applications, some EOs has been reported to cause side effects such as poisoning, asthma, contact dermatitis, headache, increased bleeding, eye-irritation, neurotoxicity, genotoxicity, and immunotoxicity. The majority of EOs, in high enough doses, will cause toxic effects. Furthermore, toxic effects may occur following ingestion or dermal exposure. Many EOs, including lavender, *Nepeta cataria* (catnip) and *Melissa officinalis* (lemon balm), have been investigated with tests demonstrating that EOs show toxicity at very low concentrations. It is supposed that one of the primary mechanisms of cytotoxicity is membrane damage, similar to that seen in bacteria and yeasts [226,227,228]. EOs also contain complex compounds that are very photosensitive and susceptible to degradation. This requires better storage conditions to prevent the component oil concentrations from decreasing and helps to maintain the primary grade essential oil with the least amount of spoilage. Another challenge for the rational exploitation of EOs by relevant industries is their quality control, as well as the legislation texts regarding their applications [228].

## 7. Antisense Antimicrobial Therapeutics

Antisense therapeutics are a biotechnological-based form of antibiotic therapy using chemical analogues of short, single-stranded oligomers that mimic the structure of RNA or DNA, and bind to specific, complementary RNA in a target organism. Antisense antimicrobial therapeutics therefore act by silencing expression of specific genes [229]. Antisense technology has been developed for many years to combat viral, parasitic and bacterial infections. Antisense technology has been used as antibacterial agents and they has been tested against most MDR clinical isolates including carbapenem-resistant *Escherichia coli*, extended-spectrum beta-lactamase *Klebsiella pneumoniae*, New Delhi metallo-beta-lactamase-1 carrying *Klebsiella pneumoniae*, and MDR *Salmonella enterica*. In this review, we’ll focus on the oligonucleotide therapy as an efficient approach to reduce the resistance of bacteria to antibiotic treatment [230].

### 7.1. Mechanism of Action

Basically, the main function of antibiotics is to directly inhibit the growth and/or viability of bacteria by targeting conserved pathways such as DNA replication, cell wall synthesis, protein synthesis and metabolism. Contrariwise, by using antisense technology as therapeutic agents in microorganisms, antisense agents bind to complementary target mRNA and silence translation or promote degradation of the targeted mRNA. Several oligonucleotide-based platforms have been developed, the two primary approaches for silencing gene expression include antisense oligonucleotides and short interfering RNAs. Both possess complementary sequences to their target mRNA. Hybridization of the oligonucleotide inhibits translation of the corresponding gene via degradation of the target mRNA or obstruction of ribosomal binding [229,231,232,233]. Currently, the main chemical structures that have been designed for antisense biotechnology include phosphorothioates, locked nucleic acids, peptide nucleic acids, and phosphorodiamidate morpholino oligomers [229,233]. The most important concern for antisense therapeutics has been the delivery of synthetic antisense oligomers into the bacterial cytoplasm that requires the attachment of another compound that can penetrate the bacterial cell wall. Recent studies have used antisense oligomers coupled by cell-penetrating peptides or a sequence of alternating cationic and non-polar amino acids because the cell walls of bacteria are nearly impenetrable by high-molecular weight oligomers [229,234,235].

### 7.2. Efficacy

In February 2017, the WHO published its first ever list of antibiotic-resistant “priority pathogens”—a catalogue of 12 families of bacteria that pose the greatest threat to human health [236]. These bacteria have become resistant to a large number of antibiotics, including carbapenems and third-generation cephalosporins [5]. As a solution to the problem, antisense therapeutics offer the potential for gene sequence-specific therapies that can target a large range of pathways in bacteria, are rapid to design, and are more adaptive to resistance than conventional molecule antibiotics [237]. Antisense oligonucleotides and short interfering RNAs are an alternative strategy to design gene-specific oligomers that can specifically target any single pathogen. This new approach nearly eliminates or significantly reduces the time required for the discovery of new antibacterials and broadens the range of potentially available targets to any gene with a known base sequence in any bacterium [229].

### 7.3. Applications in Medicine

Extended-spectrum β-lactamase production is one of the main resistance mechanisms in *Enterobacteriaceae*. Recent studies have demonstrated that the oligonucleotide therapy is an efficient approach to reduce the resistance of bacteria to antibiotic treatments [238,239]. Lipid oligonucleotides have been proven to be an efficient strategy in both delivering oligonucleotide sequences into prokaryotic cells and decrease the minimum inhibitory concentration of resistant bacteria to the third-generation cephalosporin, ceftriaxone [5]. The strategy of targeting antibiotic resistance genes aims to knock down the expression of the antibiotic resistance, which would restore susceptibility to an approved antibiotic that would be co-administered with the oligomers. Several studies have revealed the benefits of antisense antimicrobials on MDR bacteria [230,240,241]. Antisense therapeutics targeting the NDM-1 gene restored susceptibility to carbapenems in species of Gram-negative pathogens, including *Escherichia coli*, *Acinetobacter baumannii*, and *Pseudomonas aeruginosa* [230]. The same was observed when targeting the aac(6′)-Ib gene, which confers resistance to aminoglycosides in many Gram-negative clinical isolates [229,237,242,243]. There are currently no approved oligonucleotide drugs on the market for the treatment of bacterial infections; however, there are a some of antisense therapies in clinical trials for the treatment of viral infections. Fomivirsen, a phosphorothioate oligodeoxynucleotide, is the only approved antisense therapeutic that targets viral infection [229,231,237,244]. To address the urgent need for the development of new strategies for the discovery of new antimicrobials, there is still a need for more experimental studies with real clinical trials in order to successfully use them in the treatment of human infectious diseases.

## 8. Faecal Microbiota Transplant

A microbiota, or “microbial flora”, is a group of microorganisms (bacteria, viruses, fungi, or yeasts) living in a specific environment. The human body is home to different microbiota located on the skin, oral cavity, respiratory tract, genital tract or intestine. A human being contains more bacteria than human cells [245]. The intestinal microbiota is defined as the most abundant microbiota in the human body with more than 100,000 billion bacteria (1 to 2 kg of bacteria). It is now considered as an organ in its own right, providing various physiological functions essential to man. Under the influence of several phenomena, the balance between the “good” and “bad” bacteria of the microbiota can break down; this imbalance, known as dysbiosis, is the cause of various diseases of varying severity. Fecal transplantation (also called fecal transplantation or fecal bacteriotherapy) is then a possible therapeutic solution [245]. Antibiotics play a major role in these phenomena by altering the diversity of the populations of the intestinal microbiota (and in particular the barrier effect that they exert) and by increasing the intestinal densities of resistant bacteria. The intestinal microbiota is at the heart of the phenomenon of antibiotic resistance, being both the reservoir of antibiotic-resistant enterobacteria and the “collateral victim” of antibiotic use, which selects these bacteria [245]. The set of resistance genes present in the gut microbiota is called the gut resistome. It includes the endogenous or “resident” resistome composed of the bacteria of the host’s resident microbiota, and the exogenous or variable resistome, composed of resistance genes contributed by bacteria in transit through the microbiota [246]. Classically, the resident resistome includes chromosomal resistance genes not associated with mobile structures. Conversely, the variable resistome is often associated with mobile structures (such as plasmids or transposons), which can be exchanged with the host’s resident bacteria [247]. Based on the restoration of a new intestinal microbiota from a healthy donor [248]. Fecal microbiota transplantation (FMT) has been used since the fourth century in traditional Chinese medicine for the treatment of patients suffering from severe diarrhea and for the treatment of food poisoning. Indeed, this technique was first described 1700 years ago in the Chinese medical manual “Handy Therapy of Emergencies” by the Chinese physician Ge Hong [249]. Then in the 16th century, Li Shizhen described several ways of preparing the stool for preparation for fecal transplantation such as fermented fecal solution, fresh fecal suspensions, as well as dry fecal preparations of dry stool [249]. Later in the 17th century, FMT was used under the name of “transfaunation” in veterinary medicine administered orally or rectally for the treatment of diarrhea, the treatment of rumen acidosis and the treatment of bacillary dysentery or shigellosis in farm animals, of livestock [250]. In 1958, the American surgeon Ben Eiseman recognized the effectiveness of the technique of FMT by the vaginal route for the treatment of pseudomembranous colitis following a successful study on four patients treated with colonic stool (enema) from a healthy donor [250]. In 1983, the effectiveness of the treatment of recurrent *Clostridium difficile* infections by FMT was reported by Dr. Anna Schwan in the journal “*The Lancet*” [251].

### 8.1. Applications in Medicine

Digestive decontamination consists of the oral and/or parenteral administration of antibiotics in order to eradicate pathogenic and/or multi-resistant bacteria, while sparing commensal bacteria (in particular anaerobic bacteria) as much as possible [252]. Digestive decontamination has so far been used most often to eliminate enterobacteria (resistant or not) in neutropenic subjects or multi-resistant bacteria responsible for epidemics in intensive care units [247]. Emerging highly resistant bacterial infections and, in particular, carbapenemase-producing *Enterobacteriaceae*, are a growing public health problem. Their management in hospitals, regulated by a report of the French High Council for Public Health, includes measures of reinforced isolation, cohorting and screening of contacts, which are excessively cumbersome, costly and difficult to apply [247]. Antibiotic management of these colonizations is not only harmful, but totally unnecessary. Two clinical cases, one in the Netherlands, the second in France, have, on the other hand, shown the efficacy of FMT in cases of colonization by *Escherichia coli* and carbapenemase-producing *Klebsiella pneumoniae* [253]. Many studies are underway to explore this treatment option [247,254]. Stripling et al. reported, in a case report, the application of FMT to a heart–kidney transplant patient suffering from vancomycin-resistant enterococcal colonization, concomitant with recurrent *Clostridium difficile* colitis [255]. Transplantation resulted in clinical cure of the *Clostridium difficile* infection and a decrease in the proportion of vancomycin-resistant enterococci isolates [256]. The infusion of fecal preparation from a healthy donor into a patient’s gastrointestinal tract has been proposed as a novel therapeutic approach to modulate diseases associated with pathological imbalances within the resident microbiota, termed dysbiosis [257]. It has been used to treat intestinal diseases such as IBD and *Clostridium difficile* infection, but no reports are yet available on its role in treating MRSA enteritis. Here, we reported five cases of MSRA enterocolitis cured by FMT combined with vancomycin [258]. For the first time, we report changes in the stool microbiota during the therapeutic process [257].

### 8.2. Limitations

Side effects associated with fecal transplantation: results from a FMT meta-analysis concerning feces infusing from a healthy donor into the gut of a sick person to restore the balance of their gut flora [259]. This procedure has been evaluated mainly to treat recurrent *Clostridium difficile* infections. The scientific literature is dominated by small case studies with insufficient standardization of the evaluation criteria. A review of the literature included 109 publications describing this procedure performed in 1555 patients. For 1190 of them, the indication was a recurrence of *Clostridium difficile* infection [257]. For the others, fecal transplantation was performed mostly to treat chronic inflammatory bowel disease (IBD). The technique has also been investigated in irritable bowel syndrome, metabolic syndrome and constipation. As of October 2014, three randomized controlled trials could be counted: one in metabolic syndrome, one in ulcerative colitis and one for *Clostridium difficile* infection. Methods of administration were specified for 946 patients. The majority (more than 500) were transplanted by colonoscopy. It was performed by enema for more than 156 patients and by nasogastric tube for 133. This sub-analysis focuses on the 1190 patients who received a fecal transplant for recurrent *Clostridium difficile infection*. One case of fecal perforation occurred during fecal transplantation, requiring colectomy [257,260]. Two cases of norovirus contamination were reported, as well as four cases of Gram-negative bacteremia, two of which resulted in death. Among the patients also suffering from IMiC, in six cases the fecal transplantation performed to treat the *Clostridium difficile* infection was followed by an exacerbation of the inflammatory disease. In this indication, about ten febrile episodes and about one hundred mild gastrointestinal side effects were reported (nausea, abdominal pain, constipation, etc.). In addition to the deaths that occurred after bacteremia, three others were recorded. For one of the patients, the investigators considered that the link between the septic shock leading to death and the fecal transplantation was “not clear”. For another patient, it was the sedation delivered to perform the colonoscopy that was at fault. A third patient died of septic shock. Under general anesthesia, he regurgitated the fecal matter introduced by endoscopy and then fell victim to inhalation pneumonia. Other deaths that occurred in the longer term have been reported without a link to the transplant being established. For the 226 patients treated for IMiC, the most common side effect was a febrile episode (14 cases). The two authors noted six aggravations of inflammatory disease and 28 gastrointestinal side effects: flatulence, vomiting, and abdominal pain [257]. Overall, the authors consider the literature on fecal transplantation “of poor quality” and the current recommendations “insufficient” to allow practitioners to be properly informed. Based on their analysis of the literature, they consider the procedure “generally safe”, while pointing out the reporting of several cases of serious adverse events. Nevertheless, without a comparator, it is not possible to determine how much of these side effects are related to the initial health condition [257].

## 9. Quorum-Sensing Inhibitors

Quorum sensing (QS) is a cell-based communication system that controls the gene expression under the control of cell population density [261]. It controls the expression of various proteins at particular cell densities. The bacterial cells produce small hormone-like molecules known as acyl homoserine lactone (AHL) molecules, and play a role in the signaling system, hence them being known as signaling molecules [262]. These molecules are received by another bacteria as part of their signaling system, known as QS. Bacterial cells contain many different proteins, one of them AHL synthase, produces AHL molecules that are excreted outside of the cell at particular concentrations and bind with QS receptors [262]. These receptors initiate the transcription of various genes. QS controls various important features in bacteria: virulence, motility, growth, biofilm, proteases, and elastase, among others making it a pilar in most bacterial features [263]. This signaling molecule is composed of two major components: one homoserine lactone ring and a fatty acid sidechain, also known as an acyl chain [264]. Anti-virulence drugs can be used alone or in combination with antibiotics and their target, unlike antibiotics, is not to kill the bacteria but to prevent or inhibit its capacity to cause an infection [265]. Their high specificity towards the pathogen explains less adverse reaction on the host microbiota and a weaker selective pressure for drug resistance [266].

### 9.1. Pseudomonas Aeruginosa as a Model Organism

*Pseudomonas aeruginosa* is an important nosocomial pathogen. The severity of the infection is aggravated by the high resistance of this pathogen to antibiotics, as it is intrinsically resistant to most antibiotics and also acquires antibiotic resistance very easily via horizontal gene transfer. This virulence is caused by an array of virulence factors: rhamnolipids, hydrogen cyanide, pyocyanin, proteases, siderophore, and biofilms. Most of them are secreted and positively controlled by QS [267]. The possibility of inhibiting QS in *Pseudomonas aeruginosa* would be a way to shut down most of its virulence factors. *Pseudomonas aeruginosa* has 4 QS systems, each one relying on a different signal molecule: 3-oxo-C12-HSL for LasR, C4-HSL for RhlR, PQS for PsR and IQS for AmbBCDE [267]. 3-oxo-C12-HSL positively controls the expression of the other QS systems, which is why it has been the main target for the identification for an anti-virulence drug. The criteria for the selection of efficient QS inhibitors are ≤50% inhibition in bioluminescence emission and ≤20% reduction in growth. Nitrosamine, an anthelminthic was found to repress the expression of QS-regulated genes, therefore affecting the emission of pyocyanin, rhamnolipids, elastase, surface motility and biofilm formation in vitro [268]. However, mouse models of lung infection by *Pseudomonas aeruginosa* treated by nitrosamine failed, most likely because of poor solubility and bioavailability of the molecule [269]. The target of anti-virulence therapies is the las QS system and showed, in cystic fibrosis caused by *Pseudomonas aeruginosa,* promising results; however, the susceptibility of nitrosamine is variable. Another QS inhibitor in trial is clofoctol which was found to affect the growth concentration of *Pseudomonas aeruginosa* as well as secreted virulence factors and swarming motility. Other targets to QS inhibitors include *Vibrio fischeri*, *Escherichia coli*, *Staphylococcus aureus* and *Acinetobacter baumannii*, as novel therapeutic applications [270].

### 9.2. Limitations

It has been proven that resistance towards QS therapies could arise from the mutation of genes encoding efflux pumps. In silico models could be a potential solution to this developing resistance [271]. QS therapy has been shown to impact human health, side effects include the modulation of metabolic activity of the microbiota [272], but also acts as chemoattractants for neutrophils. Moreover, it was proven to induce pro-inflammatory and pro-apoptotic responses in eukaryotic cells [273]. This could lead to an exacerbation of pro-inflammatory reactions after the use of QS inhibitors. Moreover, the final results of QS therapy seem to depended on a number of causes including the immune status of the infected patient, pathogenic potential of the strain of the bacteria and environmental conditions prevailing during the infection [274]. The target of anti-virulence therapies is already known and characterized. Screening of commercial drug libraries allows to accelerate finding molecules with good pharmacological properties that will help in solving the growing problem of multidrug-resistant bacteria.

## 10. Conclusions

A new era is beginning today with the long-awaited growing research of novel anti-infectious approaches against MDR bacteria. These therapeutic approaches are definably a potent non-antibiotic option to curtail the huge increase in antibiotic resistant bacteria that we are facing nowadays. It cannot be denied that mechanisms of action of these therapies are complicated to elaborate and that clinical trials are not complete enough to provide valid and concise information. However, all efforts made in making these treatments as a clinical routine must be done with caution taking in consideration the efficacy and safety of each therapy. In the end, we are convinced that the knowledge and mastery of resistance patterns and mechanisms of action will allow clinicians to increasingly drive antimicrobial treatment towards an individualized and precise medical approach.

## Figures and Tables

**Figure 1 antibiotics-11-01826-f001:**
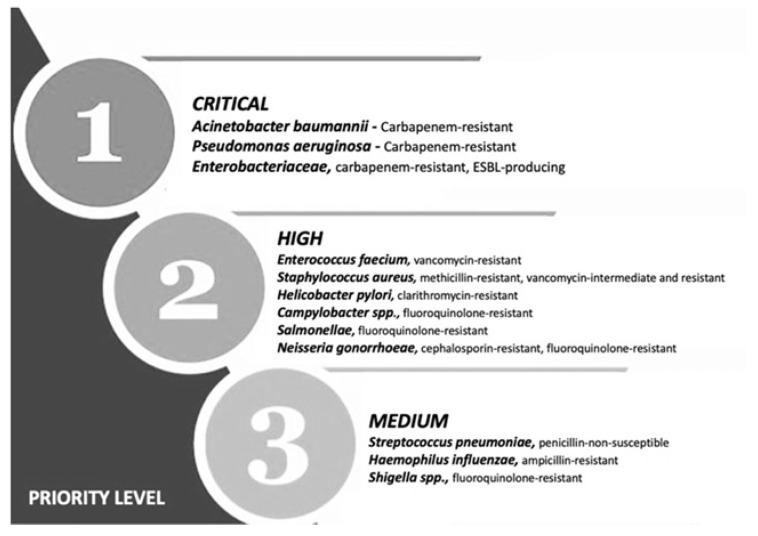
Priority list for development of new antibiotics according to the World Health Organization. Adapted from (Zyman A; et al., 2022) [6].

**Figure 2 antibiotics-11-01826-f002:**
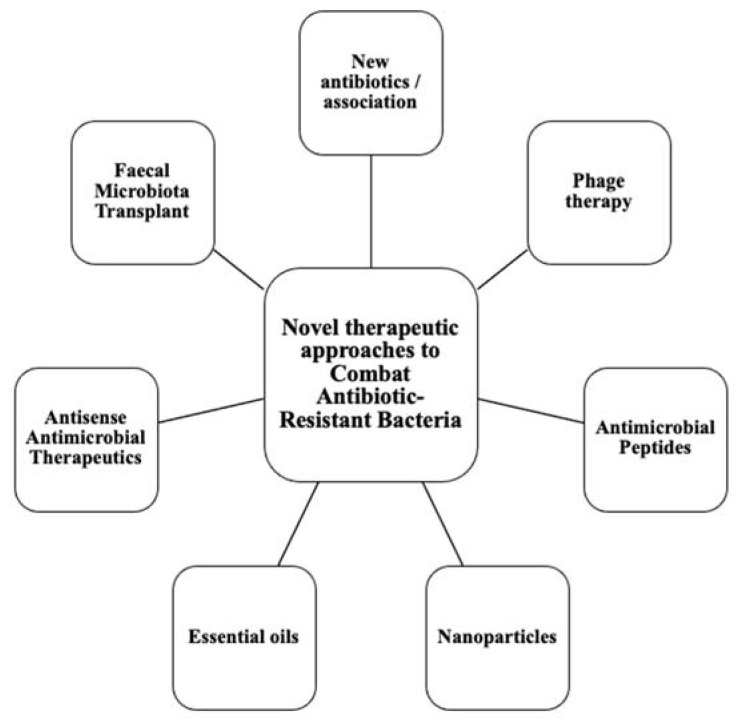
Alternatives therapeutic approaches to conventional antibiotics.

**Table 1 antibiotics-11-01826-t001:** Main advantages of bacteriophages for infections control.

Criteria	Advantages
Specificity to bacteria	Highly specific to bacteria (by specific and targeted endolysin mode of action) [79,80] Have no effect on the human host microbiota [80,81]
Effect on the immune system	Circumvent the dysbiosis and subsequent overgrowth of pathogenic species often associated with antibiotic treatment [80]
Resistance	Phage mixture minimizes the likelihood that bacteria will acquire phage resistance and kill their bacterial host quickly [82]
Effectivity on bacterial biofilms	Eradicate biofilms due to the presence of EPS-degrading enzymes like endolysins and depolymerase in their tails [79,83]
Dose	Harmless entities showing no ill effects to eukaryotic cells even at high titers (targeted therapy) [84,85,86] Capacity to naturally control bacterial populations (self-dosing property) [86,87]
Genetics	Genetic exchange between phages rarely happens [80,88]
Environmental impact	Rapid elimination from the environment [79]

**Table 2 antibiotics-11-01826-t002:** List of references of the main applications of phage therapy against multidrug-resistant bacteria classified according to the WHO priorities.

Priority	Pathogen Species	Antibiotic-Resistant Bacterium	References
Critical	*Acinetobacter baumannii*,	Carbapenem-resistant	[96,97,98,99,100,101,102]
	*Pseudomonas aeruginosa*,	Carbapenem-resistant	[103,104,105]
	*Enterobacteriaceae:*		
	*Escherichia coli*	ESBL-producing Carbapenem-resistant,	[106,107,108]
	*Klebsiella pneumoniae*,	Multidrug-resistant Carbapenem-resistant	[109,110,111,112] [112,113]
	*Enterobacter* spp.,	Carbapenem-resistant	[114,115]
High	*Enterococcus faecium*,	Vancomycin-resistant	[116,117,118]
	*Staphylococcus aureus*,	Methicillin-resistant, vancomycin-resistant,	[119,120,121,122]
	*Helicobacter pylori*,	Clarithromycin-resistant,	[123,124,125]
	*Campylobacter* spp.,	Fluoroquinolone-resistant,	[126,127]
	*Salmonellae*	Fluoroquinolone-resistant,	[128,129,130]
	*Neisseria gonorrhoeae*,	Cephalosporin-resistant, Fluoroquinolone-resistant,	[131,132]
Medium	*Streptococcus pneumoniae*,	penicillin-non-susceptible,	[133,134]
	*Haemophilus influenzae*,	Ampicillin-resistant,	[135,136,137]
	*Shigella* spp.,	Fluoroquinolone-resistant,	[138,139,140]

**Table 3 antibiotics-11-01826-t003:** Limits of application of phage therapy in modern human medicine.

	Limits of Application of Phage Therapy	References
1	Dosage of bacteriophages, duration of treatment and routes of administration, is poorly controlled, involving the safety and effectiveness of treatment.	[68]
2	Inability to replicate the in vitro results in the actual situations.	[141]
3	Results of experimentation in small animal models does not consistently translate into clinical success, just as in vitro phage activity often fails to correlate with in vivo efficacy.	[136,142,143]
4	As same as antibiotics, bacteria also develop resistance to phages by specific defense mechanisms.	[80,144]
5	Phages display a short circulation time due to clearance by the spleen.	[145]
6	Bacterial remnants in the lysate produced from mass production of phages are difficult to be completely eliminated, leading to health risks.	[144,145]
7	Strain specificity of phages hinders mass production.	[76]
8	Possibility to contribute to the antimicrobial resistance development through transduction “phage conversion”.	[81]
9	Key mechanisms that may allow the prediction of in vivo pharmacokinetics and dynamics linked to therapeutic outcome have not yet been fully elucidated.	[68]
10	Physicochemical properties of phages in vivo are not fully understood.	[68]
11	Lack of regulatory approval for human use.	[84]

## Data Availability

The data supporting the conclusions are included within the manuscript.

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
