# Peer review of "Alternatives Therapeutic Approaches to Conventional Antibiotics: Advantages, Limitations and Potential Application in Medicine"

_antibiotics, 2022, doi:10.3390/antibiotics11121826_

Round 1

Reviewer 1 Report

The submitted review represents a comprehensive literature study that identifies alternative therapeutic approaches to conventional antibiotics. The manuscript is well documented, and the information is accessible to even less familiar readers. The abstract accurately summarises the review's content, and recent literature was considered.

Text comments:

L.62: erase increased "mortality."

L.83: does the reference (8) corresponds to written text?

L.133: the meaning "seems reasonable". Clinical results? Risk-benefit?

L. 146: "as its oral formulation was discontinued after its disappointing failure and poor outcomes in clinical studies" Does It mean Oral Bioavailability?

L. 145-150: reference, please.

L. 365: reference (s). please

L.425: erase "with a specific target," please (it is repeated)

L.629 – 630: reference, please.

Author Response

Response to Reviewer 1 Comments

Thank you very much for the reviewers comments concerning our manuscript entitled ”Alternatives therapeutic approaches to conventional antibiotics: advantages, limitations and potential application in medicine". (ID: antibiotics-1958387).

Your comments are all valuable and very helpful for revising and improving our paper. We have studied comments carefully and have made corrections accordingly, which we hope will meet with approval. 

Our responses to your comments and questions are marked in red below: 

Point 1. L.62: erase increased "mortality."

Response 1:

  • L63 - “increased mortality” has been removed.

Point 2. L.83: does the reference (8) corresponds to written text?

Response 2:

  • Reference (8) corresponded only to the antimicrobial activity of Plazomicin.
  • 88: Reference (13,14) has been added to better justify the written text.

Point 3. L.133: the meaning "seems reasonable". Clinical results? Risk-benefit?

Response 3:

  • L152–181 has been rephrased and references (31-39) has been added.

Point 4. L. 146: "as its oral formulation was discontinued after its disappointing failure and poor outcomes in clinical studies" Does It mean Oral Bioavailability?

Response 4:

  • L189-191: Exactly; According to the references (42,43) added to the text, Oral bioavailability of Eravacycline is estimated at 28 % (range 26–32 %).

Point 5. L. 145-150: reference, please.

Response 5:

  • L191: Reference (42) has been added to substantiate the role of Eravacycline in the spread of carbapenems and reference (43) have been added to explain the limitations of the modes of administration of Eravacycline.

Point 6. L. 365: reference (s). please

Response 6:

  • L423: Reference (165) has been added to explain the impact of a small modification in the amino acids on the function of the synthesized protein.
  • L425: Reference (166) was added to address the last sentence of this paragraph.

Point 7. L. 425: erase "with a specific target," please (it is repeated)

Response 7:

  • "with a specific target," has been removed from line L. 484

Point 8.  L. 629 – 630: reference, please.

Response 8:

  • L678-684: This part of manuscript has been rephrased and references (236) and (237) have been added.

Reviewer 2 Report

Thanks for allowing me to review the manuscript. Generally, it is an interesting review with novel knowledge for readers. However, many aspects still required thorough revision.

  1. Typo and grammatical mistakes, like "convectional", might be "conventional", and "2.4 New association" might be "New combination"? I am not sure, etc.
  2. It seems that the manuscript was written by many authors and then synthesized together. Therefore, many sections have a similar and redundant beginning; for example, 4.1, lines 311-314; 6.2, lines 545-547; 7, lines 595-597; 7.2, lines 628-636; lines 640-642; etc;
  3. In the previous section, the abbreviation has already been introduced, but the full name may occur in the later sections, like and not limited to MDR, etc. It should be checked thoroughly;
  4. The species' name should be in iliac styles, like Enterobacteroaceae; please check carefully.
  5. Reference should be added immediately following the citation contents. In some sentences, the author said, "many studies……", but only with one reference. 
  6. Regarding the marketed antibiotics, like plazomicin, eravacycline, and new beta-lactam combinations, it is recommended to add information about their action mechanisms, epidemiology data of in vitro activity, clinical evidence supporting their effectiveness, and PK/PD profiles. The author focuses so much on chemical structures instead of the practice aspect of these approved new antibiotics in the manuscript. 
  7. Faecal microbiota transplant might not be suitable in the present review, as it is majorly used for Clostridium difficile infections. 
  8. In the abstract, the authors mentioned the present review would highlight the importance of the combination therapy approach, but no such information is in the body of the manuscript.  
  9. The conclusion could not refect the review's main points; it is recommended to rewrite this part.

Author Response

Response to Reviewer 2 Comments

Point 1. Typo and grammatical mistakes, like "convectional", might be "conventional", and "2.4 New association" might be "New combination"? I am not sure, etc.

Response 1:

  • The word "convectional" has been deleted and replaced by the word "conventional" in the title of the manuscript and in the title of "figure 1".
  • "2.4 New association" has been replaced by "New combination antibiotic therapy". It is the most used expression in literature.

Point 2. It seems that the manuscript was written by many authors and then synthesized together. Therefore, many sections have a similar and redundant beginning; for example, 4.1, lines 311-314; 6.2, lines 545-547; 7, lines 595-597; 7.2, lines 628-636; lines 640-642; etc. 

Response 2:

  • Indeed, the manuscript was written by many authors. we have reworked and rephrased the reported parts to minimize or even avoid redundancy.

Point 3. In the previous section, the abbreviation has already been introduced, but the full name may occur in the later sections, like and not limited to MDR, etc. It should be checked thoroughly;

Response 3:

  • All abbreviations with their full name have been carefully checked. After each abbreviation, the full name no longer appears.

Point 4. The species' name should be in iliac styles, like Enterobacteroaceae; please check carefully.

Response 4:

  • All species names have been checked carefully and are written in italic style. Except for species names in references.

Point 5. Reference should be added immediately following the citation contents. In some sentences, the author said, "many studies……", but only with one reference.

Response 5:

  • For each assertion one or more references were immediately added.
  • Each time "Several studies" or "many studies" has been cited, we have enriched it with the missing references.

Point 6. Regarding the marketed antibiotics, like plazomicin, eravacycline, and new beta-lactam combinations, it is recommended to add information about their action mechanisms, epidemiology data of in vitro activity, clinical evidence supporting their effectiveness, and PK/PD profiles. The author focuses so much on chemical structures instead of the practice aspect of these approved new antibiotics in the manuscript.

Response 6:

  • We tried to briefly describe the classes of antibiotics their structural characteristics, while highlighting their spectrum of action and field of clinical application. Part 2.2 (L.152-181); 2.3 (215-227); 2.4.1 (244-257); and 2.4.2 (277-282) have been reworked and references have been added to better respond to your comment.

Point 7. Faecal microbiota transplant might not be suitable in the present review, as it is majorly used for Clostridium difficile infections.

Response 7:

  • Although current evidence considers FMT to be a generally safe therapeutic method with few adverse events, the long-term outcomes of FMT have not been fully elucidated. Therefore, establishing the periodicity and duration of regular follow-up after FMT to monitor clinical efficacy and long-term adverse events are other critical issues. In the future, we look forward to personalized FMT for different patients and conditions based on various hosts and diseases. 

Point 8: In the abstract, the authors mentioned the present review would highlight the importance of the combination therapy approach, but no such information is in the body of the manuscript.

Response 8:

  • Several references have been cited and others have been added to the manuscript to highlight the importance of the "combination therapy approach". We cite as an example:
    • Refs 2 and 107: "phage therapy in combination with antibiotics"
    • Ref. 196: "nanoparticles combination to antibiotics / nanobiotics"
    • Ref 224: "Combined effect of antibiotics and essential oils against Campylobacter multidrug resistant strains and their biofilm formation"
    • Ref 238: "Antisense oligonucleotides in combination with nanoparticles against methicillin-resistant S. aureus"
    • Ref 258: "five cases of MSRA enterocolitis cured by FMT combined with vancomycin"
    • Ref 265 and 266: "Anti-virulence drugs in combination with antibiotics"

Point 9: The conclusion could not refect the review's main points; it is recommended to rewrite this part.

Response 9:

  • The conclusion has been rewritten in order to reflect the main points raised in our manuscript.

Reviewer 3 Report

This outstanding and comprehensive review describes the current state of affairs with respect to antimicrobial resistance, focusing on the 12 bacteria that comprise the World Health Organization's priority list. Recently approved antibiotics and a number of promising investigational treatments are discussed. Throughout, the drug and/or treatment mechanisms that provide activity against ESBL and other multidrug resistance platforms are discussed in detail, along with toxicity and the current and/or proposed place in therapy. The paper is current, well designed, and a great read. 

Author Response

Response to Reviewer 3 Comments

Thank you very much for your comments that perfectly summarizes the objectives of our manuscript entitled: Alternatives therapeutic approaches to conventional antibiotics: advantages, limitations and potential application in medicine" (ID: antibiotics-1958387).

Hopefully, that review provides instructive points for our healthcare professionals.

Special thanks to you for your positive feedback.

Round 2

Reviewer 2 Report

Nice revision, and I have no other comments.